# Relational Coordination at the Primary–Secondary Care Interface: Insights from a Cross-Sectional Survey in the South Tyrolean Healthcare System

**DOI:** 10.3390/ijerph21040425

**Published:** 2024-03-30

**Authors:** Christian J. Wiedermann, Verena Barbieri, Adolf Engl, Giuliano Piccoliori

**Affiliations:** 1Institute of General Practice and Public Health, Claudiana—College of Health Professions, 39100 Bolzano, Italy; 2Department of Public Health, Medical Decision Making and Health Technology Assessment, University of Health Sciences, Medical Informatics and Technology, 6060 Hall in Tirol, Austria

**Keywords:** relational coordination, healthcare teamwork, communication dynamics, professional group interaction, cross-sectional survey

## Abstract

Understanding the dynamics of teamwork and communication among healthcare professionals is crucial in the face of evolving healthcare challenges. This study assessed relational coordination among healthcare professionals in the South Tyrolean healthcare system in Italy, focusing on communication and teamwork dynamics in a cross-sectional survey. Using the validated Relational Coordination Survey (RCS) instrument and 525 completed online responses, the questionnaire aimed to understand the implications of different levels of relational coordination ratings by general practitioners, hospital physicians, nurses, and administrative personnel (response rate 26%). The demographics of the participants revealed a predominance of female professionals (64%), with an average age of 50 and 18 years of service. The resulting RCS scores varied significantly across professional groups, with nurses reporting the highest within-group scores, indicating moderate coordination, and administrators reporting the lowest scores, reflecting areas of weak coordination. Between-group relational coordination was generally perceived as weak across professional groups, with the least weakness observed between general practitioners and nurses. German or Italian language and health district affiliation emerged as significant factors influencing relational coordination ratings, highlighting the need for differentiated understanding and strategies in multilingual and diverse regional settings. Assessments of interdisciplinary feedback and referral practices highlight the variation in teamwork and communication weaknesses and underscore the need for targeted interventions to improve relational coordination. This study provides insights into the complexity of relational dynamics in health care settings. This suggests that improving relational coordination through tailored strategies could significantly improve team effectiveness, quality of patient care, and overall system efficiency.

## 1. Introduction

In an era of unprecedented global health challenges, the post-COVID-19 landscape has further underscored the critical importance of a robust and responsive health workforce [1]. As the backbone of health systems worldwide, health professionals—from doctors and nurses to allied health workers—face myriad challenges exacerbated by the pandemic. These challenges include coping with an increased workload, rapidly evolving disease patterns, and the need for innovative care delivery models. The pandemic has not only highlighted the vulnerability of health systems but also the indispensable role of the health workforce in navigating these complexities, ensuring continuity of care, and adapting to the evolving health needs of populations. Therefore, understanding and addressing the intricacies of health workforce dynamics is paramount to supporting resilient health systems capable of responding effectively to current and future global health challenges [2].

Relational coordination (RC) has emerged as a central framework for understanding and improving the functioning of health systems. RC, as defined by Gittell [3], is a mutually reinforcing process of interaction between communication and relationships for task integration. RC consists of key components, such as frequent, timely, accurate, and problem-solving communication, coupled with relationships characterized by shared goals, shared knowledge, and mutual respect. The relevance of RC in healthcare is particularly pronounced given the inherent complexity of the sector, interdependence of tasks, and critical need for seamless collaboration among diverse professionals. Effective RC in healthcare can lead to improved quality of care, greater efficiency, and higher patient satisfaction [4]. In a healthcare complex network, where multidisciplinary teams address diverse health issues, the principles of RC offer a critical perspective to enhance communication and relationships, leading to more effective and cohesive care delivery.

The health workforce faces significant challenges that impact care quality and accessibility. Urban–rural maldistribution leads to unequal access, necessitating strategies for equitable healthcare distribution [5]. Furthermore, a shortage of primary and preventive care workers compromises early intervention and chronic disease management, which is critical for maintaining population health [6]. Additionally, demographic, epidemiological, social, and technological shifts demand a responsive and adaptable workforce [7]. Addressing these challenges is crucial for a robust and future-ready healthcare system. The relevance of RC lies in its potential to improve health care services by improving teamwork among professionals. Teams with strong RC are better able to provide integrated, adaptive, and effective care [8]. This is particularly important in the post-COVID-19 era, when health systems require teams that can coordinate effectively under pressure and adapt to rapidly changing circumstances.

The literature reveals a gap in understanding the internal dynamics of healthcare teams, particularly RC, and their impact on workforce stability and quality of care, especially when resources are scarce [9]. There is also a critical need for customized solutions that recognize the unique challenges of different healthcare environments and avoid one-size-fits-all strategies to address specific local needs and dynamics effectively.

The primary objective of this study aims to investigate the state of RC among healthcare professionals in South Tyrol, Italy, focusing on understanding internal team dynamics and their direct effects on healthcare delivery and workforce stability. This study also seeks to contribute to the international conversation on health workforce challenges and solutions, particularly in the evolving post-COVID-19 landscape [10]. While it offers a context-specific understanding and solutions for South Tyrol, it also provides insights that are applicable in broader contexts. The implications of RC among healthcare professionals and their broader significance are discussed, along with suggestions for practical solutions to enhance healthcare workforce stability.

## 2. Materials and Methods

### 2.1. Study Design and Setting

This cross-sectional observational study was designed to assess the status of RC among physicians, nurses, and administrators in South Tyrol’s public health service. While this study lays the groundwork for understanding the broader impacts of RC, specific analyses related to job satisfaction and intention to stay will be addressed in a subsequent dedicated paper.

Similar to the English health system [11], the Italian National Health Service is publicly funded. Within Italy, South Tyrol is an autonomous province with a population of 532,616 (31 December 2022), made up of the following language groups: German (62.3%), Italian (23.4%), Ladin (4.1%), and other (10.2%) (as of the 2011 census) [12]. The current shortage of health professionals is even more serious than in the rest of Italy [13] given the need for bilingual health professionals [14,15]. Additionally, the healthcare sector lags behind in terms of digitalization for similar reasons [16,17]. This area, located in northern Italy, is characterized by a healthcare system that operates within a setting that presents distinct challenges and opportunities for healthcare delivery and professional collaboration. Understanding RC in this context provides valuable perspectives on workforce dynamics and health care quality in diverse regional settings.

### 2.2. Participants

Participants included healthcare professionals actively engaged in healthcare delivery by the South Tyrolean Health Authority (SABES–ASAA), specifically general practitioners (GPs), hospital physicians, nurses, and administrative personnel directly involved in ambulatory patient care. The participant selection of this study was focused on professionals directly involved in outpatient care, including those with dual roles in outpatient and inpatient settings, to specifically examine the relational dynamics at the primary–secondary care interface. The exclusion criteria were non-clinical personnel who were not directly involved in patient care and professionals working outside the public healthcare system.

Recruitment for this study was conducted through targeted e-mail invitations to specific groups of healthcare professionals across South Tyrol. The SABES–ASAA invited employed specialist physicians, nurses, and administrative personnel, whereas the Institute of General Practice and Public Health extended invitations to GPs contracted with the Health Authority. To maximize participation and ensure comprehensive coverage, two reminders were sent two and four weeks after the initial invitation.

The survey was executed as an anonymous online questionnaire (Appendix A) utilizing SoSci Survey Software, version 3.2.46. Conducted from 28 August 2023 to 2 October 2023, the period aimed to capture a representative snapshot of current attitudes and experiences. Participation in the survey was voluntary and informed consent was provided, and participants were informed about this study’s purpose, confidentiality measures, and their right to withdraw at any time. Completion of the RCS items of the questionnaire was mandatory.

The demographic data collected included age, gender, years of experience, job role, and any additional relevant factors that might influence perceptions of RC and job satisfaction. In compliance with ethical guidelines, health district data were collected only from GPs, hospital physicians, and nurses, with administrative staff exempted to safeguard privacy.

The SABES–ASAA is organized into four anonymized health districts: Health District 1, Health District 2, Health District 3, and Health District 4. This anonymization strategy is used to prevent public discussion of quality ratings that could distort perceptions of the health services provided by these districts. Each district has its own characteristics and capacities, designed to efficiently meet the health needs of the local population. One serves as the main hub, providing a wide range of specialized medical services. Two others, though smaller, are well equipped to provide comprehensive healthcare services, focusing on the health needs of the community. One specializes in specific areas of medicine to meet the needs of both residents and visitors seeking specific treatments. This organizational structure ensures that health services are accessible and tailored to the diverse needs of the South Tyrolean population, while maintaining confidentiality in quality assessment discussions.

### 2.3. Questionnaire

The core of the data collection was the Relational Coordination Survey (RCS), which is a validated instrument designed to measure RC within organizations [3]. The RCS assesses seven dimensions of communication and relationships that are crucial for understanding the dynamics within healthcare teams. These dimensions include the following.

Frequency of Communication: How often team members communicate.Timeliness of Communication: The promptness of information exchange.Accuracy of Information: The reliability and correctness of shared information.Problem-Solving Communication: The focus on resolving issues collaboratively rather than blaming.Shared Goals: The extent to which team members have common objectives.Shared Knowledge: The mutual understanding of each member’s roles and tasks.Mutual Respect: The degree of reciprocal respect and appreciation among team members.

Each dimension is rated using a 5-point Likert scale, which provides a quantitative measure of RC within the healthcare environment [4,18].

### 2.4. Data Analysis

The reliability of the RCS was assessed using Cronbach’s alpha to ensure internal consistency across seven dimensions: frequency, timeliness, accuracy, problem solving, shared goals, shared knowledge, and mutual respect. The overall and individual alphas for different professional groups (GPs, specialists/hospital physicians, administrative staff, and nursing staff) were calculated. Weighting factors were introduced to address potential response bias due to the varying response rates across professional groups. The cut-off points for weak, moderate, and strong RC ties within and between roles were based on the norms of previously collected RC scores collected between 2012 and 2015 [19].

Descriptive statistics were used to summarize the survey data. This included calculating means, medians, standard deviations, and ranges for continuous variables, such as the RCS scores for each dimension, and frequencies and percentages for categorical variables, such as gender, language, and health district. The overall mean age of the participants and the distribution of demographic characteristics were also reported to contextualize the findings.

Spearman correlation and Kruskal–Wallis tests were used to assess the relationships between the RCS dimensions and various factors, including age, sex, language of the questionnaire, health district, years of service, adherence to referral criteria, and feedback on referrals. The strength of RC within and between workgroups was categorized as weak, moderate, or strong, based on specific score ranges.

All statistical analyses were performed using the IBM SPSS Statistics for Windows (version 25.0; IBM Corp., Armonk, NY, USA).

## 3. Results

Of the 856 total visits to the questionnaire platform, 145 individuals did not agree to the privacy policy and were therefore not included. In addition, 16 administrative staff participants (10% of this group) were excluded from the survey because they indicated that they had no direct or indirect professional relationship with the other professional groups. 170 participants did not respond to the RCS. Table 1 shows the response statistics by profession and language preference. Participation across different healthcare professional groups reflects varying levels of engagement and response rates, with the nursing staff showing the highest rate of participation. A total of 186 participants discontinued the survey during the course of completing the RCS, leaving 525 responses that had a complete RCS section and were considered evaluable. A smaller group of 14 individuals (2.7%) exited the survey during the remaining questions but were still deemed evaluable for analysis.

The reliability of the RCS was tested. The overall Cronbach’s alpha for the seven items was 0.835, indicating a high level of internal consistency. Reliability remained robust when broken down by professional group: GPs had a Cronbach’s alpha of 0.884, hospital physicians 0.857, administrative personnel 0.776, and nurses 0.788.

### 3.1. Demographic and Professional Characteristics

In the demographic analysis of the survey participants, the majority were female (63.9%), with an overall mean age of 50 years with a standard deviation (SD) of 8.5 and mean length of service of 18 years (SD 10.5). The proportion of German-speaking participants was 57.3%.

Professional roles showed different demographic data, with nurses having the highest percentage of female participants (87.4%) and the highest mean (M) and standard deviation (SD) number of years of service (M 19, SD 11.0). GPs had a balanced gender distribution but the highest percentage of missing data (9.8%). Hospital physicians and administrative staff had similar mean ages and years of service, with a slight male predominance in both the categories. Non-administrative staff participants were distributed across four health districts, with the largest representation from Health District 1 (42.1%) (Table 2).

Participants from Health District 3 and Health District 4 had the highest percentage of German speakers (78.8%), whereas Health District 1 had the lowest (38.5%). The average age and length of service did not vary significantly between districts as well as gender distribution.

### 3.2. Relational Coordination Scoring

The RC scores within the SABES–ASAA healthcare setting revealed varying degrees of perceived collaboration quality, both within and between professional groups (Table 3).

Internally, nurses reported the best scores for RC within their group (4.25), which were rated as ‘moderate’. To a lesser extent, GPs rated their internal within-group RC as ‘moderate’, with a score of 4.17. Hospital physicians and administrative staff scored 3.92 and 3.22, respectively. These two groups fall into the ‘weak’ category. Externally, RC between professional groups is generally perceived as weaker. GPs rated the quality of their RC with hospital physicians (3.30) and administrative personnel (2.93) as ‘weak’ and with nurses (3.97) as ‘moderate’. Hospital doctors rated their external RC with GPs (3.26), administrative personnel (3.15), and nurses (3.35) as ‘weak’. Similarly, administrators rated their external RC as ‘weak’ across the board (3.11 to 3.19). Nurses rated the RC scores for administrators as ‘moderate’ (3.58), but for GPs and hospital doctors also as ‘weak’ (3.44 and 3.27, respectively). Notably, none of the external RC scores reached the ‘strong’ category.

The weighted overall scores, considering the response rates, suggest ‘weak’ RC for the work with GPs and administration staff and ‘moderate’ RC for the work with hospital physicians and nurses. This classification does not differ from the unweighted classification.

The strongest intergroup coordination was rated by GPs for the RC between GPs and nurses (3.97), which is close to being classified as ‘strong’. The weakest intergroup coordination was actually rated by GPs for the RC between GPs and administrative staff (2.93).

#### 3.2.1. Within-Group Relational Coordination by Dimension

Table 4 details the internal RC scores by dimension within various healthcare professional groups.

GPs demonstrated moderate RC within their group, with all dimensions scoring above the moderate threshold, except for the ‘Frequency of communication’, ‘Accuracy of information’, and ‘Shared goals’ dimensions.

Hospital physicians reported a weak overall RC, with individual scores revealing a relative weakness in the ‘Shared goals’, ‘Shared knowledge’, ‘Accuracy of information’, ‘Frequency of communication’, ‘Problem-solving communication’, and ‘Timeliness of communication’ dimensions. Only ‘Mutual respect’ scored above the moderate threshold.

Administrative staff showed the weakest internal RC, with all seven dimensions falling below the threshold for moderate coordination. The ‘Frequency of communication’, ‘Shared goals’, and ‘Shared knowledge’ dimensions are areas of concern.

While nurses have better overall internal RC, they do not have any weak dimensions, which underlines their more cohesive internal communication and collaboration practices.

The overall weighted scores across all professional groups indicated that the ‘Frequency of communication’, ‘Shared goals’, and ‘Shared knowledge’ dimensions were most frequently identified as weak.

#### 3.2.2. Between-Group Relational Coordination by Dimension

Table 5 illustrates the external RC scores of different professional groups within the healthcare setting by dimension, with particular emphasis on dimensions where the scores were classified as ‘weak’ (<3.5).

In the analysis of between-group RC, the overall weighted score indicates a general trend towards weak external coordination between these professional groups. This finding is consistent across most dimensions.

The dimension of ‘Frequency of communication’ reveals particularly low scores across all groups, with the lowest reported by administrative staff, followed by hospital physicians, and GPs. Nurses rated this dimension slightly higher, but it still falls within the ‘weak’ category. ‘Timeliness of communication’ also received weak ratings, particularly from hospital physicians. In contrast, ‘Accuracy of information’ was rated comparatively higher, with administrative staff reporting the highest score. However, hospital physicians rated this dimension lower, indicating variability in the perceived precision of information exchange.

‘Problem-solving communication’ achieved the highest weighted overall score among the dimensions, bordering on ‘moderate’ coordination. This dimension was rated highest by GPs and lowest by hospital physicians. The dimensions of ‘Shared goals’ and ‘Shared knowledge’ received weak ratings overall, with administrative personnel reporting particularly low scores. ‘Mutual respect’ stands out with a ‘moderate’ overall score, highlighting a level of professional esteem, particularly as rated by hospital physicians and administrative staff.

Appendix A illustrates the RC scores among healthcare professional groups across seven dimensions, shedding light on the dynamics of internal and external coordination. The ‘Frequency of communication’ dimension is generally rated ‘weak’ across all groups, with GPs (3.19), hospital physicians (3.12), and administrators (2.45. Nurses, however, rated it moderately (3.57) for GPs, suggesting a relatively better frequency in their interactions. ‘Timeliness of communication’ scores are weak to moderate, with GPs rating it strong (4.11) for nurses. Other groups, including hospital physicians (3.32) and administrators (3.32), indicate a moderate level of ‘Timeliness in communication’. The ‘Accuracy of information’ dimension sees a mix of weak to moderate ratings. GPs rated it as moderate for both hospital doctors (3.55) and nurses (3.67), while administrators rated it moderately across all groups (3.63). ‘Problem-solving communication’ is rated as moderate across most groups, with GPs rating it strong (4.27) for nurses. ‘Shared goals’ is generally rated weak, except for GPs rating it strong for nurses (4.09). The ‘Shared Knowledge’ dimension received weak to moderate ratings, with GPs rating it as strong for nurses (4.40) but weak for administrators (2.95). The lowest score was reported by administrators for their group (2.71). ‘Mutual Respect’ is the strongest dimension overall, with moderate to strong ratings. Both GPs (4.46) and hospital physicians (4.17) rated it ‘strong’ for nurses, indicating high levels of respect among these groups.

The overall weighted scores reflect a trend towards moderate coordination, with the highest scores in ‘Mutual respect’ and the lowest in ‘Frequency of communication’.

#### 3.2.3. Referral Compliance and Feedback among General Practitioners

Approximately 20% of GPs believe that their compliance with the referral criteria does not exceed 50%. Furthermore, a significant majority of GPs (63.4%) seldom or never received feedback from specialists.

Regarding hospital physicians, nearly half (48.4%) perceived over 30% of referrals from GPs as being inappropriately prioritized. Feedback on such referrals is also lacking, with 57.6% of hospital doctors indicating that they rarely or never provide referrals. Moreover, almost one-third of hospital doctors (29.9%) rated the quality of clinical questions posed by GPs as very poor or poor.

#### 3.2.4. Impact of Demographic and Professional Factors on Relational Coordination

The overall RC score did not differ by rater gender (male, female, not specified). Similarly, neither the within-group nor the between-group scores in both the weighted and unweighted versions differed significantly between genders in the overall scores, nor did the subgroups by profession. Sub-scores per professional group differed significantly for hospitalists (male 3.72, female 3.48, not reported 3.75; *p* = 0.001), administrative staff (male 3.16, female 3.36, not reported 2.98; *p* = 0.001), and nurses (male 3.60, female 3.91, not reported 3.53; *p* < 0.001), but not for GPs. ‘Frequency of communication’ (male 3.16, female 3.35, not reported 3.15; *p* = 0.001), ‘Accuracy of information’ (male 3.48, female 3.59, not reported 3.22; *p* = 0.01), and ‘Timeliness of communication’ (male 3.42, female 3.46, not reported 3.13; *p* = 0.03) were the dimensions that differed by gender.

Further, the overall as well as the between-group RC scores differed between languages of the raters (German 3.58, Italian 3.45; *p* = 0.003, and German 3.42, Italian 3.23; *p* < 0.001, respectively), while there was not found any difference for the within-group score. The weighted scores did not differ at all. The only professional group-depending sub-score that was different between languages was the score of the GP’s (German 3.64, Italian 3.23; *p* < 0.001). The dimensions ‘Timeliness of communication’ (German 3.58, Italian 3.21; *p* < 0.001), ‘Shared goals’ (German 3.53, Italian 3.41; *p* = 0.028) and ‘Mutual respect’ (German 4.06, Italian 3.80; *p* < 0.001) also differed between languages.

Finally, the largest differences were found between health districts, under the restriction, that administrative staff is not included in this analysis. The overall RC (Health District 1 3.49, Health District 2 3.71, Health District 3 3.67, Health District 4 3.49), as well as the within-group RC (Health District 1 4.11; Health District 2 4.11; Health District 3 4.15; Health District 4 4.06), the between-group RC (Health District 1 3.28; Health District 2 3.57; Health District 3 3.51; Health District 4 3.30), and the RC of the for subgroups GP’s, hospital practitioners and nurses differed highly significant between health districts (*p* < 0.001, each). The subdimensions ‘Frequency of communication’ (Health District 1 3.32; Health District 2 3.37; Health District 3 3.34; Health District 4 3.28; *p* < 0.001), ‘Timeliness of communication’ (Health District 1, 3.30; Health District 2, 3.68; Health District 3, 3.70; Health District 4, 3.45; *p* = 0.003), ‘Accuracy of communication’ (Health District 1, 3.52; Health District 2, 3.72; Health District 3, 3.70; Health District 4, 3.44; *p* < 0.001), ’Shared goals’ (Health District 1, 3.43; Health District 2, 3.69; Health District 3, 3.74; Health District 4, 3.41; *p* = 0.04) and ‘Shared knowledge’ (Health District 1, 3.45; Health District 2, 3.52; Health District 3, 3.28; Health District 4, 3.43; *p* < 0.001) were also significantly different between health districts, while ‘Problem-solving communication’, and ‘Mutual respect’ did not differ. Weighted RC scores were significantly different for the overall RC score (Health District 1, 3.43; Health District 2, 3.67; Health District 3, 3.63; Health District 4, 3.42; *p* = 0.007) and the within-group RC score (Health District 1, 4.03; Health District 2, 4.06; Health District 3, 4.09; Health District 4, 3.95; *p* < 0.001), while the between-group weighted score did not differ.

Appendix A presents a comprehensive analysis of the differences in RC scores between various metric demographic and professional factors, as well as feedback practices. Findings indicate that age and years of service do not significantly correlate with overall RC scores. However, notable correlations are observed in feedback practices and perceptions. Specifically, hospital physicians’ perceptions of inappropriate referral priority demonstrate a negative correlation with both overall and between-group RC scores. Similarly, the quality of clinical questions posed by GPs shows a significant negative correlation with RC. The dimension of ‘Frequency of communication’ positively correlates with overall RC, suggesting its importance in relational dynamics. Other dimensions, including timeliness, accuracy, problem solving, shared goals, and mutual respect, also exhibit significant correlations, particularly with hospital physicians’ perceptions.

## 4. Discussion

### 4.1. Overview of Relational Coordination in the South Tyrolean Health Authority

This study examined the status of RC within the SABES–ASAA. The findings reveal certain areas within the RC framework where SABES–ASAA functions acceptably, as well as others with notable weaknesses that could potentially affect the overall effectiveness and stability of the healthcare workforce.

Overall, differences in the region’s main language groups, German and Italian, and in health district affiliation were significantly associated with differences in RC scores, with RC scoring by German-speaking professionals better than Italian-speakers, and professionals in smaller health districts scoring RC better than in larger health districts. Language has a consistently significant effect on several dimensions of RC, particularly between different professional groups.

The overall RC score was similar for all genders. However, when looking at the specific professional groups other than GPs, i.e., hospitalists, administrators, and nurses, there were differences in ‘Frequency of communication’, ‘Accuracy of information’, and ‘Timeliness of communication’, based on gender.

The GPs’ low adherence to referral criteria and their low perception of feedback on referrals were significantly correlated with low RC scores. This correlation confirms that lower compliance and less frequent feedback are associated with lower RC. Notably, hospital doctors’ assessment of the quality of clinical questions asked by their colleagues was significantly correlated with overall RC and intergroup coordination, suggesting that perceived quality of communication is integral to collaborative practice.

For hospital doctors, gender showed a significant correlation with internal RC, suggesting that gender dynamics may influence collaborative interactions within this group.

For administrators, a negative correlation was observed between their perception of inappropriate referral priorities and RC, suggesting that administrative processes may influence their engagement in collaborative practice.

Nurses, who typically rated their internal coordination highly, also showed significant correlations with gender and health district, suggesting that these factors influenced their perceptions of RC. However, ‘Mutual respect’ is most strongly correlated with RC, highlighting the importance of professional respect within nursing teams.

### 4.2. Relational Coordination’s Role in Healthcare Effectiveness and Workforce Stability

Recognizing the complex nature of collaboration among health workforce groups, this study has attempted to focus on RC as the primary variable of interest. However, the important role that bureaucratic procedures, health system organization, methods of payment for services, and training programs play in shaping these collaborative dynamics is acknowledged. While these factors were beyond the scope of our current analysis, their influence on interprofessional collaboration is undeniable and warrants comprehensive investigation.

RC fundamentally promotes quality communication and relationships between professionals, enabling them to coordinate their work effectively, especially under conditions of interdependence, uncertainty, and time constraints [20]. In healthcare settings, such as SABES–ASAA, where the complexity of patient care often requires seamless collaboration across specialties and functions [21], the importance of RC cannot be overstated. Effective RC not only enhances the immediate working environment by improving communication and teamwork but also has a significant impact on patient outcomes, staff satisfaction, and the overall efficiency of healthcare delivery [22,23]. Weaknesses in RC can lead to fragmented care, inefficiencies, increased staff stress, and potentially higher turnover rates [24]. These internal dynamics are critical not only to the quality of patient care but also to the broader operational stability and effectiveness of health services.

The findings of this study, which point to specific areas where RC could be improved within the SABES–ASAA, have significant relevance beyond the immediate context. Healthcare systems worldwide face similar issues: increasing demand for services, the need for multidisciplinary care, and challenges to staff retention and satisfaction [25]. The findings of this study contribute to the wider discourse on the stability and efficiency of the healthcare workforce. This is particularly relevant in the post-COVID-19 era, when pressures on health systems have increased, making the need for effective coordination and communication more important than ever [26].

Demographic factors play an important role in shaping RC dynamics in healthcare settings. Differences in age, gender, professional experience, and specific roles within the healthcare team can significantly influence how individuals communicate and collaborate [27]. Increasing age had a negative impact on RC in this SABES–ASAA study. Age can affect communication styles, adaptability to change, and the use of technology [28]. Younger staff members may be more open to new technologies and communication methods, whereas older staff may prefer traditional face-to-face interactions. A generational divide in communication preferences and working styles can lead to misunderstandings and reduced coordination efficiency.

Gender dynamics can influence communication styles, leadership roles, and team interactions [29]. Cultural and societal norms may influence individuals’ assertiveness or receptiveness in collaborative settings. Gender bias or stereotypes were prevalent among hospital doctors. They may hinder open and effective communication and contribute to a lack of mutual respect and shared understanding within teams [30].

‘Problem-solving communication’ and ‘Shared goals’ emerged as critical dimensions, with demographic and professional factors playing important roles. Experienced staff tend to have a deeper understanding of the healthcare system and better problem-solving skills [8]. However, they may also be more resistant to changes in established practices. Differences in experience levels can lead to differences in communication and decision-making processes [21].

Different roles bring about different perspectives, knowledge bases, and priorities. For example, doctors, nurses, and administrators may have different focuses and approaches to patient care [27]. Without a clear understanding and respect for the contribution of each role, there may be a lack of shared goals and knowledge. This can manifest itself in the fragmentation and lack of coordinated problem solving.

Weak RC has significant implications for workforce stability and efficiency, impacting job satisfaction, burnout, and turnover rates [31]. Poor RC can lead to increased stress and decreased intention to stay among healthcare professionals, contributing to a destabilized workforce. High turnover rates not only disrupt patient care but also impose financial and knowledge losses [32]. Weak RC makes recruitment more challenging, as negative workplace reputation deters potential high-quality candidates. Therefore, addressing RC weaknesses is crucial for reducing burnout and turnover, enhancing job satisfaction, and creating an attractive and efficient work environment.

To improve RC within the SABES–ASAA and address its identified weaknesses, a diverse approach to targeted interventions is proposed. First, training programs are considered a cornerstone for fostering better communication and teamwork skills [33]. The vision is to conduct workshops and training sessions that not only focus on effective communication strategies and conflict resolution but also emphasize the importance of empathy and interdisciplinary teamwork in the healthcare setting. Ensuring that staff not only attend these sessions but also actively apply the principles learned in their daily interactions requires both commitment and cultural change within the organization. Simultaneously, policy changes aimed at creating a collaborative and supportive working environment are considered essential [8]. This could include the introduction of policies that encourage regular interdisciplinary meetings, creating clear and effective communication protocols, and establishing recognition programs for successful teamwork.

Technological solutions are also being developed to improve information sharing and coordination among healthcare professionals. Importantly, mobile phones for internal and external communication are not yet uniformly available in SABES–ASAA health districts and may partly explain the observed differences in RC between health districts. Upgrading or implementing new real-time communication platforms and electronic health record systems could revolutionize the way information is shared and discussed within a team. However, the introduction of new technologies is often met with resistance [34], and concerns regarding usability, privacy, and security must be fully addressed.

The way to make these changes is not only about the interventions themselves but also about how they are implemented [35]. Leadership buy-in at all levels is critical for supporting these changes and providing necessary resources [36]. Involving staff in the design and implementation processes can lead to greater acceptance and better alignment of solutions with real needs. Finally, consideration of the unique linguistic, cultural, and organizational context of the SABES–ASAA will be key to ensuring that these interventions are positively received and effectively integrated into daily practice. By addressing these considerations and focusing on these targeted interventions, the SABES–ASAA can make significant strides in addressing the weaknesses of RC, paving the way for a more collaborative, efficient, and resilient healthcare workforce.

### 4.3. Limitations, Implications and Future Directions

The unique cultural and linguistic landscape of South Tyrol, characterized by multilingualism and specific health care challenges, may initially seem to limit the generalizability of the findings. However, this specific context provides a valuable model for examining relational coordination in settings of comparable sociocultural diversity, which are not uncommon worldwide. Many global regions face similar challenges of linguistic diversity and multiculturalism within health care systems, making the findings of our study potentially applicable and valuable beyond the confines of South Tyrol. By highlighting the need for contextually adapted strategies to improve healthcare teamwork and communication, this study invites a broader exploration of how diverse healthcare settings can implement these findings. This approach underscores the importance of comparative studies to adapt and generalize principles of relational coordination across sociocultural landscapes.

In an effort to comprehensively account for and mitigate potential biases arising from linguistic and cultural diversity within South Tyrol, this study strategically included participants from both peripheral and central health districts. This approach ensured a diverse representation of healthcare professionals across different linguistic and cultural backgrounds, facilitating a balanced analysis of relational coordination.

A notable limitation was the exclusion of hospital nurses and non-medical health professionals other than administrative staff from this study. Additionally, despite attempts at weighting, varying participation rates between different professional groups and health districts may have affected the representativeness of the results. While RC items were derived from validated instruments and showed reliability in our sample, other survey items were not validated.

The findings of this study and the proposed interventions for the SABES–ASAA are in line with the global health workforce challenges. The international migration of health workers highlights the need for robust RC to manage diverse and often overstretched teams [37]. Global health emergencies, such as COVID-19, underscore the need for adaptable, well-coordinated teams, while the evolving nature of healthcare delivery models requires a workforce that is skilled in technology and multidisciplinary collaboration. The proposed strategies, which focus on communication, training, and technology, offer globally applicable lessons and point to more resilient, efficient, and adaptable health systems.

Future research should strive for inclusivity, involving a wider range of healthcare professionals, to provide a more comprehensive picture of RC across the healthcare spectrum. Longitudinal studies would be particularly beneficial for tracking the long-term impact and sustainability of interventions aimed at improving RC. A detailed analysis of job satisfaction and intention to stay is also crucial, and will be reported separately. In addition, comparative studies across different healthcare settings could validate and extend our findings, whereas intervention-based research can provide concrete evidence on the effectiveness of specific strategies.

## 5. Conclusions

This study explored RC within a regional Italian health service characterized by linguistic and cultural diversity, providing insights into the dynamics of communication and collaboration among health professionals. Notable findings include the weakness of the RC in several areas, significant associations between RC scores and factors such as language, local health district affiliation, and gender dynamics, particularly within hospital physicians and administrators. Specifically, linguistic group membership significantly impacted several RC dimensions, underscoring the importance of effective communication in healthcare. In addition, this study found an association between GPs’ non-adherence to referral criteria, their perception of low referral feedback, and RC scores. Similarly, hospitalists’ assessment of the quality of clinical questions posed by GPs was related to overall and between-group RC. These relationships highlight the critical role of communication quality and feedback mechanisms in collaborative practice. This study also found that nurses, who generally rated their internal coordination highly, showed significant correlations with gender and health district, indicating that these demographic factors influenced their perceptions of RC. This is particularly evident in the strong correlation between mutual respect and RC, emphasizing the importance of professional respect within nursing teams. The important role of RC in promoting quality communication and relationships in health care settings is stressed, especially under conditions of interdependence and complexity. The results indicate specific areas where RC could be improved within SABES-ASAA, with broader implications for health systems facing similar challenges worldwide. This study points to the need for targeted interventions to improve the communication, teamwork, and collaborative environments that are essential for building a resilient health workforce and delivering quality patient care.

## Figures and Tables

**Table 1 ijerph-21-00425-t001:** Response statistics by professional area and language.

Professional Area	Surveys Sent	German Speaking		Italian Speaking		Total Participants		Response Rate	
		Platform Access and Consent	RCS Completed	Platform Access and Consent	RCS Completed	Platform Access and Consent	RCS Completed	Platform Access and Consent	RCS Completed
GPs	289	70	56	37	26	107	82	37%	28%
Hospital Physicians	1290	148	99	163	132	311	231	24%	18%
Administrative Personnel	155	21	6	36	8	57	14	37%	9%
Nurses	320	168	140	68	58	236	198	74%	62%
Overall Totals	2054	407	301	304	224	711	525	35%	26%

Abbreviations: RCS, Relational Coordination Survey; GP, general practitioner.

**Table 2 ijerph-21-00425-t002:** Demographic and professional characteristics of survey participants.

Variable	Age (M ± SD years)	Years in Service (M ± SD)	Gender			German Speaking (%)	Health District (Non-Administrative Staff Participants)				
			Female (%)	Male (%)	Missing (%)		Bolzano (%)	Merano (%)	Bressanone (%)	Brunico (%)	Missing (%)
Total	50 ± 8.5	18 ± 10.5	63.9	27.2	8.9	57.3	43.2	27.6	12.9	10.2	6.1
Gender											
Female	48 ± 7.6	17 ± 10.6	-	-	-	59.4	48.1	27.5	11.4	11.7	1.2
Male	50 ± 10.2	18 ± 10.3	-	-	-	53.8	39.7	31.9	19.1	9.2	0.0
No declaration	49 ± 8.9	18 ± 10.7	-	-	-	57.9	47.4	36.8	10.5	5.3	0.0
Health district (only non-administrative staff participants)											
Health District 1	49 ± 8.4	17 ± 10.0	70.6	25.3	4.1	38.5	-	-	-	-	
Health District 2	50 ± 9.4	17 ± 11.3	56.1	40.9	3.0	78.8	-	-	-	-	
Health District 3	49 ± 8.1	17 ± 10.9	73.0	25.0	1.9	78.8	-	-	-	-	
Health District 4	49 ± 8.1	18 ± 10.2	63.1	3.9	5.0	70.2	-	-	-	-	
Professional group											
GPs	50 ± 11.0	14 ± 10.5	47.6	42.7	9.8	68.3	40.2	18.3	19.5	12.2	9.8
Hospital physicians	49 ± 8.9	18 ± 9.3	48.5	39.8	11.7	42.9	43.7	22.9	13.4	13.0	6.9
Administrative personnel	49 ± 21.1	21 ± 16.2	78.6	14.3	7.1	42.9	n.d.	n.d.	n.d.	n.d.	n.d.
Nurses	49 ± 6.8	19 ± 11.0	87.4	7.1	5.5	70.7	43.9	36.9	9.6	6.1	3.5

Abbreviations: M, mean; SD, standard deviation; n.d., no data.

**Table 3 ijerph-21-00425-t003:** Relational coordination scores among healthcare professional groups.

		Ratings of ^1^					
		GPs	Hospital Physicians	Administrative Personnel	Nurses	External	Overall
Ratings by ^1^	GPs	4.17	3.30	2.93	3.97	3.40	3.59
	Hospital physicians	3.26	3.92	3.15	3.35	3.25	3.42
	Administrators	3.11	3.19	3.22	3.16	3.16	3.17
	Nurses	3.44	3.27	3.58	4.25	3.43	3.64
	Total	3.46	3.56	3.28	3.78	3.29	3.52
	Total (weighted)	3.40	3.67	3.19	3.57	3.34	3.46

^1^ Relational coordination within workgroups scoring: weak < 4.1, moderate 4.1–4.6, strong > 4.6; between workgroup scoring: weak < 3.5, moderate 3.5–4.0, strong > 4.0 [19]. Abbreviations: GPs, general practitioners.

**Table 4 ijerph-21-00425-t004:** Mean relational coordination scores by dimension within healthcare professional groups.

RC Dimensions	Professional Groups				Overall (Weighted)
	GPs	Hospital Physicians	Administrators	Nurses	
Within-group RC ^1^	4.17	3.92	3.22	4.25	3.95
Frequency of communication	3.65	3.93	3.07	4.09	3.85
Timeliness of communication	4.28	3.91	3.36	4.32	3.98
Accuracy of information	4.05	3.85	3.43	4.26	3.91
Problem-solving communication	4.27	3.90	3.29	4.20	3.95
Shared Goal	4.09	3.81	3.07	4.20	3.86
Shared knowledge	4.40	3.83	2.71	4.28	3.89
Mutual respect	4.46	4.17	3.64	4.42	3.92

^1^ RC, relational coordination within workgroups scoring: weak < 4.1, moderate 4.1–4.6, and strong > 4.6 [19].

**Table 5 ijerph-21-00425-t005:** Mean relational coordination scores by dimension of healthcare professional groups.

RC Dimensions	Professional Groups RC with				Overall (Weighted)
	GPs	Hospital Physicians	Administrators	Nurses	
Between-group RC ^1^	3.40	3.25	3.16	3.43	3.4
Frequency of communication	3.03	2.84	2.24	3.39	2.91
Timeliness of communication	3.26	3.12	3.31	3.25	3.17
Accuracy of information	3.55	3.18	3.69	3.46	3.32
Problem-solving communication	3.59	3.48	3.43	3.52	3.50
Shared Goal	3.31	3.13	3.10	3.32	3.18
Shared knowledge	3.33	3.11	2.48	3.28	3.12
Mutual respect	3.69	3.91	3.80	3.38	3.86

^1^ RC, relational coordination between workgroups scoring: weak < 3.5; moderate 3.5–4.0, strong > 4.0 [19].

## Data Availability

The data presented in this study are available upon request from the corresponding author. The data were not publicly available because the survey had politically sensitive content.

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
