# Peer review of "Relational Coordination at the Primary–Secondary Care Interface: Insights from a Cross-Sectional Survey in the South Tyrolean Healthcare System"

_ijerph, 2024, doi:10.3390/ijerph21040425_

Round 1

Reviewer 1 Report

Comments and Suggestions for Authors

11. Although as a principle the subject of coordination between various healthcare workers and how it influences quality of care is of tremendous significance to the scientific community, in my opinion, the study presented (despite being very thorough) has a very serious limitation – it addresses the issues of a limited region of Tyrol with very specific cultural and ethnic background.  It can not be considered representative even for Italy as a whole, and even less so for Europe. As a result, the general interest it is small.

22.There are many factors influencing collaboration between groups of healthcare workers which are not discussed in the manuscript – bureaucratic procedures, organization of health system, methods of refunding for services provided, training programs and so on.

33. I do not see a reason for which the hospital nurses and administrative stuff (non-clinic) were excluded  - they are an integrated part of the inner workings of the health system. This introduces in the study a strong possibility of bias.

44.      The title is misleading since there is not actual analysis of the stability and performance of the workforce – there is no correlation presented between the scores and the performance of care, the leaving of jobs or anything like this.

55.      The questionnaire should be presented as a supplementary material

Comments on the Quality of English Language

Minor editing needed.

Reviewer 2 Report

Comments and Suggestions for Authors

This paper aims to investigate the type and grade of Relational coordination among a cohort population of healthcare professionals in South Tyrol, between Italy and Swiss/Austria. 

An overall evaluation is positive, survey is well structured and clear. Data seems to be very interesting; the most relevant information is well described in the dogma «Similes cum similibus congregantur». As example, nursers are down to work with nurses (RC score<4), with a weak relation with Administrators. So I agree with the sentence: ".In the analysis of between-group RC, the overall weighted score indicates a general trend towards weak external coordination between these professional groups.."

You wrote that Sud Tirol is an indipendent region, so did you consider as a bias the social skills of this North Italy population? 

Did you share these data with the survived health professionals? 

How can a Director avoid the weak relationship among health and administrative workers? For example, the share of informations is a part of security of care, how do you propose to improve it?

Round 2

Reviewer 1 Report

Comments and Suggestions for Authors

I believe that the authors have carefully addressed all my suggestions and provided adequate responses. They made sufficient changes for the manuscript to be published as it is. 

Comments on the Quality of English Language

Minor editing needed